# Does the Use of Intraoperative Neuromonitoring during Thyroid and Parathyroid Surgery Reduce the Incidence of Recurrent Laryngeal Nerve Injuries? A Systematic Review and Meta-Analysis

**DOI:** 10.3390/diagnostics14090860

**Published:** 2024-04-23

**Authors:** Andrew Saxe, Mohamed Idris, Jickssa Gemechu

**Affiliations:** Department of Foundational Medical Studies, Oakland University William Beaumont School of Medicine, Rochester, MI 48309, USA; mohamedidris@oakland.edu (M.I.); gemechu@oakland.edu (J.G.)

**Keywords:** intraoperative nerve monitoring, thyroid surgery and parathyroid surgery, recurrent laryngeal nerve

## Abstract

Injury to the recurrent laryngeal nerve (RLN) can be a devastating complication of thyroid and parathyroid surgery. Intraoperative neuromonitoring (IONM) has been proposed as a method to reduce the number of RLN injuries but the data are inconsistent. We performed a meta-analysis to critically assess the data. After applying inclusion and exclusion criteria, 60 studies, including five randomized trials and eight non-randomized prospective trials, were included. A meta-analysis of all studies demonstrated an odds ratio (OR) of 0.66 (95% CI [0.56, 0.79], *p* < 0.00001) favoring IONM compared to the visual identification of the RLN in limiting permanent RLN injuries. A meta-analysis of studies employing contemporaneous controls and routine postoperative laryngoscopy to diagnose RLN injuries (considered to be the most reliable design) demonstrated an OR of 0.69 (95% CI [0.56, 0.84], *p* = 0.0003), favoring IONM. Strong consideration should be given to employing IONM when performing thyroid and parathyroid surgery.

## 1. Introduction

The recurrent laryngeal nerve (RLN) provides motor innervation to the intrinsic muscles of the larynx (except the cricothyroid) which produce phonation. Injury to a RLN can result in paresis or paralysis of the ipsilateral vocal cord. Unilateral vocal cord paralysis can produce significant changes in voice, while bilateral cord paralysis can result in asphyxiation. In addition to its motor function, branches of the RLN provide sensory innervation to the laryngeal mucosa below the level of the vocal cords. Interference with this function can lead to aspiration. Even in the absence of life-threatening complications, injuries of the RLN affect patients’ quality of life [1]. From the surgeon’s perspective, injury to the RLN is the most common reason for malpractice litigation related to thyroid surgery [2].

The anatomical relationship of the RLN makes it vulnerable to intraoperative injury. The RLN is a branch of the vagus nerve (cranial nerve X) that enters the thoracic cavity and then returns (“recurs”) to the neck to lie along the trachea in intimate proximity to the thyroid gland. The left RLN curves below and behind the aortic arch just posterolateral to the ligamentum arteriosum in the superior mediastinum, whereas the right RLN loops under the right subclavian artery at the root of the neck. Both nerves ascend lateral to the trachea and lie in the tracheoesophageal groove, posterior to the thyroid gland as it courses to the larynx. The nerve, however, has considerable variation in its course and branching pattern within the neck. Occasionally, the left RLN branches before entering the larynx and, in approximately 1% of cases, the right RLN does not follow the normal looping pattern but instead enters the neck directly superior-laterally from the vagus nerve [3]. The anatomical variation contributes to the risk of injury.

The incidence of intraoperative injury to the RLN is difficult to assess with precision for the reasons highlighted by Dionigi et al. [4]. It is often associated with transient and minimal voice disturbances so that patients are unaware of, or reluctant to report, their disability. Only a subset of reports in surgical series includes rigorous pre- and post-operative voice or vocal cord assessments. The incidence of total (permanent plus transient) RLN injuries reported in the literature we reviewed with greater than 50 nerves at risk (NAR) varied between 1.4 and 19.5% [5,6] and permanent injuries between 0–6.7% [7,8]. Extended resections for malignancy, reoperations, retrosternal goiter, and Graves’ disease are associated with a greater incidence of RLN iatrogenic injuries [9,10].

A consensus exists that the most important intraoperative maneuver to minimize risk to the RLN is visualization of the nerve early in the procedure, prior to embarking upon thyroid or parathyroid excision [11,12,13,14]. However, visual identification of the nerve may be difficult for the reasons mentioned earlier. Perhaps the most promising, yet still controversial, adjunct method for visualization alone is intraoperative neuromonitoring (IONM). Since its introduction in 1966, IONM has been promoted as offering surgeons several benefits including an enhanced RLN identification rate, a reduction in identification time, the detection of anatomic variations of the RLN, and the assessment of the postoperative function of the vocal cords. The underlying proposition of IONM is that, by applying an electrical current to the nerve and simultaneously assessing vocal cord movement, one can determine whether the nerve is intact. Several methods of IONM have been evaluated both with respect to nerve stimulation and vocal cord assessment. With respect to vocal cord assessment, the most common method is the use of electrodes incorporated into an endotracheal tube. Other methods include laryngeal palpation [15] and the trans-tracheal insertion of needle electrodes into the vocal cords [7,16,17]. Nerve stimulation, typically 0.5–1.5, A at 30 Hz delivered by bipolar electrodes, can be delivered intermittently by the surgeon (intermittent IONM [I-IONM]) or continuously (continuous IONM [C-IONM]). Currently, the most commonly used method is applying endotracheal tube surface electrodes to the mucosa of the vocal cord stimulated intermittently by bipolar electrodes conveying an electric current of 0.5–1.5 mA at 30 Hz [18,19,20]. The two types of stimulations currently used, I-IONM and C-IONM, provide unique advantages. I-IONM can be used periodically to confirm the identity of the RLN prior to performing the critical portions of the procedure. In principle, C-IONM can detect a distressed nerve and impending injury, whereas I-IONM can detect a nerve injury only after it has occurred [21].

Proponents argue that IONM enhances the surgeon’s ability to identify, and therefore protect, the RLN, especially in high-risk procedures. With the use of IONM, rates of temporary vocal cord palsy range between 0.5% [10] and 12.5% [22], and rates of permanent vocal cord paralysis range between 0% [8,17,23,24,25,26,27,28,29,30,31] and 5.8% [6] among series with a minimum of 50 NAR. However, convincing evidence for the utility of IONM is lacking due to conflicting reports. Several meta-analyses have produced contrary results, as shown in Table 1. Recognizing the inconsistent results among meta-analyses, Sanabria et al. published a review of these meta-analyses and highlighted the shortcomings and deficiencies of existing meta-analyses [32]. Among the deficiencies identified were the use of a single database, incorporating studies that do not have control groups, and the use of relative summary statistics rather than absolute summary statistics. In an attempt to answer the question, “Does IONM reduce the incidence of RLN injury?” we performed a systematic review and meta-analysis which addressed most of those deficiencies.

## 2. Materials and Methods

We conducted a systematic literature review according to the guidelines of the Preferred Reporting Items for Systematic Reviews and Meta-Analyses (PRISMA). PubMed, EMBASE, Cochrane, Web of Science, and ISRCTN registry databases were queried for all human studies addressing the efficacy of RLN monitoring during thyroid surgery and parathyroid surgeries. Searches were updated by PubMed automated recurrent searches. This study has no PROSPERO registration number and no registered protocol.

Literature search strategies were developed using medical subject headings (MeSH) combined with operators “AND” or “OR” and text words appropriate for the respective databases. Examples of keywords are, “thyroid surgery”, “thyroidectomy”, “parathyroidectomy”, “nerve monitoring”, “recurrent laryngeal nerve”, “recurrent laryngeal nerve injury”, “vocal cord paralysis”, and “neuromonitoring”. We employed Covidence software [Systematic Review Software, Veritas Health Innovation, Melbourne, Australia] to house citations and track progress in screening and reviewing citations in compliance with the PRISMA algorithm.

Data were extracted to a Sheets [Google, Mountain View, CA, USA] spreadsheet.

RevMan 5 [Cochrane Computer Program Version 5.0, Copenhagen, Denmark] was used for the statistical analysis and assessment of the risk of both study bias and publication bias. To limit selection bias, both the screening process and data extraction were undertaken independently by two different reviewers.

The primary outcome was the number of RLN injuries among patients for whom IONM was used compared to the number of injuries among patients for whom IONM was not used (No IONM). This was calculated both as a function of injuries per RLN at risk of injury (NAR) and as injuries per patient. In studies that reported intentional RLN divisions or patients with preoperative nerve dysfunction, we subtracted those from the total number of NAR to establish the number of nerves truly at risk and amenable to preservation by IONM. Other data collected included the following: date and country of study, age, gender, number of patients, type of surgical approach, extent of surgery, type of disease for which the surgery was done, equipment used, technique of assessing RLN injury, length of surgery, type of control (contemporaneous vs. historical), and study design (prospective vs. retrospective, randomized vs. nonrandomized).

Statistical tests employed Mantel–Haenszel odds ratios (ORs) using both a fixed and random model. Forest plots using a fixed model are displayed in the Figures. Because we conducted 10 separate analyses, we applied a Bonferroni correction to establish a more conservative level of ά = 0.005 for each analysis.

### Inclusion/Exclusion Criteria

Inclusion: Thyroid and parathyroid surgeries that used intraoperative neuromonitoring of the recurrent laryngeal nerve; randomized and non-randomized studies with controls; all patients regardless of age and gender; all relevant studies regardless of date; articles in English; human subjects. We included studies in which there were no nerve injuries among patients in whom IONM was and was not employed (so-called “both armed zero-event studies”) [46].

Exclusion: Studies that included patients with prior recurrent laryngeal nerve damage or vocal cord dysfunction; studies employing unconventional surgical procedures, e.g., trans-axillary endoscopic procedures; studies derived from multi-institutional databases; studies by authors that included patients previously reported upon.

## 3. Results

A flow chart of the literature selection process, including criteria for excluding studies, is shown in Figure 1. All the studies included in the final analysis are non-randomized and retrospective, except for five randomized trials and eight non-randomized prospective trials. The studies selected compared IONM plus visual nerve identification to visual nerve identification alone for the prevention of RLN injury in participants undergoing conventional thyroidectomy and parathyroidectomy. Most studies (50) used endotracheal tubes (vs. 10 studies with needle electromyography) to detect the electromyographic (EMG) signal. All of the studies are published in English, are from 17 countries, the majority from the United States and China, and have publication dates ranging from 1992 to 2022. In total, studies included 28,318 patients, with a median age of 45.7 years (range 1–93 years) and a female preponderance with 74% in the patient population. Table 2 presents the characteristics of patients, operations, and study designs included in the selected studies.

The studies displayed a great deal of heterogeneity with respect to design, surgical pathology, patient demographics, and assessment of outcome. All studies used I-IONM stimulation except those of Zhou et al. [47], Adamczewski et al. [19], and Anuwong et al. [48] where both I-IONM and C-IONM were analyzed. Surgical pathology ranged from benign to malignant neoplasms, different types of goiters, and hypothyroidism. The number of patients with benign thyroid pathologies predominated with 64% compared to 32% of malignant thyroid pathologies and only 4% of parathyroid pathologies. Overall, 52.5% of the surgical operations were total thyroidectomies, 17.3% lobectomies, 14.1% reoperations, 8.7% node dissection, 4.4% parathyroidectomies, 2% subtotal thyroidectomies, and 1.1% near-total thyroidectomies. Assessment varied with respect to reporting RLN injuries as a function of NAR or number of patients. We believe the preferred assessment uses NAR. The total number of NAR was 77,270, of which 49,204 (64%) were in the IONM group and 28,066 (36%) were in the RLN control group. Injury assessment varied from subjective voice analysis to postoperative laryngoscopy.

Because of data heterogeneity, we performed several meta-analyses stratifying the studies according to study design (randomized RCTs vs. nonrandomized RCTs), type of RLN injury (overall vs. permanent vs. combined), and assessment of RLN injury (use or not of post-operative laryngoscopy). We did not assess transient RLN paralyses separately because few studies reported those uniquely. Instead, we analyzed permanent injuries and the total number of RLN injuries.

**Table 2 diagnostics-14-00860-t002:** Characteristics of studies included in this meta-analysis.

Author [Citation]	Year	Country	Dates of Study	Type of Study	Type of Control	NAR (Patients)	Postop Laryngoscopy	Definition of Permanent Injury	Number of Surgeons	% Female Patients	% Cancer Operations	% Total Thyroidectomy
Adamczyk [19]	2015	Poland	1992–2005	R	C	120 (80)	Yes	6 months	M	100	NS	100
Agha [49]	2008	Germany	2012–2017	R	C	(59)	Yes	NS	M	61	NS	100
Akici [50]	2020	Turkey	2004–2012	R	H	(273)	Yes	6 months	S	86	0	100
Akkari [51]	2014	France	2005–2012	R	C	(90)	NS	NS	NS	75	NS	44.6
Alesina [52]	2014	Germany	2002–2014	PNR	C	1708 (1114)	Yes	6 months	M	75	9	47
Anuwong [48]	2016	Italy	2003–2007	R	H	NS	Yes	NS	S	79	19	NS
Atallah [53]	2009	France	2006–2007	R	C	421 (261)	Yes	12 months	M	77	7	60
Barczyński [54]	2009	Poland	1993–2012	PR	C	2000 (1000)	Yes	12 months	M	91	12	75
Barczyński [55]	2014	Poland	1993–2012	R	C	1326 (854)	Yes	12 months	M	81	28	39
Bonati [56]	2022	Italy	2009–2015	R	H	1212 (638)	No	NS	M	76	95	90
Brajcich [57]	2016	USA	1995–2002	R	H	1048 (627)	Yes	12 months	S	83	20	67
Brauckhoff [23]	2002	Germany	2007–2014	R	H	169 (97)	Yes	NS	M	58	77	NS
Calò [5]	2017	Italy	2002–2005	R	H	4730 (2365)	Selective	12 months	M	80	31	100
Chan [58]	2006	Hong Kong	2001–2010	R	C	1000 (639)	Yes	12 months	M	79	22	61
Chuang [59]	2013	Taiwan	2009–2012	R	H	83 (71)	No	NS	S	83	24	57
DeDanschutter [24]	2015	Netherlands	2014–2016	R	H	170 (147)	Yes	12 months	M	86	14	16
Demiryas [60]	2018	Turkey	1998–2001	R	C	370 (191)	Yes	12 months	S	9	NS	46
Dralle [16]	2004	Germany	2008–2009	PNR	C	23,349	Yes	6 months	M	78	7	69
Duclos [61]	2011	France	2013–2018	PNR	C	(686)	Yes	NS	M	78	21	76
Dudley [62]	2021	USA	2008–2016	R	C	(107)	No	3 weeks	M	58	NS	NS
Ercetin [25]	2019	Turkey	NS	PR	C	1496 (748)	Yes	12 months	M	89	NS	NS
Frattini [63]	2010	Italy	2007–2010	R	NS	304 (152)	Yes	NS	NS	56	100	100
Gremillion [64]	2012	USA	2007–2010	R	C	162 (119)	NS	NS	S	NS	NS	36
Grishaeva [65]	2022	Germany	2007–2011	R	H	2720 (1963)	Yes	6 months	M	77	0	38
Hayward [9]	2013	Australia	2012–2014	R	C	3736	NS	NS	NS	NS	18	53
Hei [66]	2016	China	1997–2016	PR	C	84 (70)	Yes	6 months	S	77	77	26
Kadakia [18]	2017	USA	2013–2016	R	H	(1418)	Yes	8 months	S	68	57	NS
Kai [67]	2017	China	1987–2008	R	C	836 (552)	Yes	6 months	M	22	25	89
Karakas [68]	2013	Germany	2014–2016	R	H	(111)	NS	NS	M	65	NS	NS
Kartal [20]	2021	Turkey	2009–2019	R	C	839 (493)	Yes	6 months	M	79	19	70
Kim [6]	2020	Korea	2011–2014	PNR	C	133 (121)	Yes	12 months	M	72	100	100
Lee [69]	2017	Australia	2008–2018	PNR	H	1583 (990)	Yes	NS	S	80	18	60
Legre [17]	2020	France	2014–2018	R	C	77 (47)	Selective	6 months	M	62	13	38
Leow [70]	2020	Singapore	2012–2017	R	C	301 (193)	Yes	6 months	M	74	25	15
Ling [71]	2020	China	2010–2016	R	C	1696 (1033)	Yes	6 months	M	73	46	76
Long [72]	2018	China	2012–2014	R	C	(435)	Yes	6 months	M	32	100	100
Mirallie [73]	2018	France	1997–2016	PNR	C	2633 (1328)	Yes	6 months	M	80	22	100
Mourad [74]	2017	USA	1997–2016	R	H	(213)	Yes	7 months	S	72	0	0
Netto [75]	2007	Brazil	2003–2006	R	H	327 (204)	Yes	3 months	M	93	58	63
Page [76]	2015	France	2001–2010	R	H	1534 (767)	Yes	12 months	M	82	NS	100
Pei [77]	2021	China	2010–2020	R	H	159 (109)	Yes	NS	M	56	47	NS
Polat [78]	2015	Turkey	2010–2012	PNR	C	178 (94)	Yes	NS	M	79	27	76
Prokopakis [30]	2013	Greece	2004–2011	R	C	121 (97)	No	4 months	M	79	87	25
Ritter [26]	2021	Israel	2001–2019	R	C	183 (113)	Yes	12 months	M	75	57	43
Robertson [79]	2004	USA	1999–2002	R	C	236 (165)	Yes	NS	M	77	33	NS
Sanguinetti [28]	2014	Italy	2012	R	C	700 (350)	Yes	NS	M	NS	NS	100
Sari [29]	2010	Turkey	2007–2009	PR	C	409 (237)	Yes	12 months	M	82	17	79
Sharif [80]	2017	Pakistan	2014–2017	R	C	400 (200)	Yes	6 months	M	NS	NS	100
Shindo [81]	2007	USA	1998–2005	R	C	1043 (684)	Yes	NS	S	NS	55	52
Sopinski [82]	2017	Poland	2014–2016	R	C	133 (80)	NS	NS	M	95	0	66
Stevens [83]	2012	USA	2004–2008	PNR	C	143 (91)	No	6 months	M	59	41	57
Teksoz [27]	2015	Turkey	2011–2012	PR	C	322 (161)	Yes	6 months	M	76	34	100
Terris [31]	2007	USA	2004–2006	R	C	176 (137)	Yes	6 months	S	NS	18	28
Thong [84]	2021	Ireland	2009–2019	PNR	H	1539 (1001)	Yes	6 months	S	81	23	55
Vasileiadis [10]	2016	Greece	2002–2012	R	C	5112 (2556)	Yes	12 months	M	79	NS	100
Witt [85]	2005	USA	1998–2003	R	C	190	NS	12 months	S	NS	NS	44
Wojtczak [8]	2017	Poland	2011–2014	R	C	105 (61)	Yes	12 months	M	87	20	85
Yarbrough [22]	2004	USA	1998–2003	R	H	151 (111)	Selective	NS	M	63	66	4.5
Zhang [7]	2020	China	2018	R	C	280 (200)	Yes	6 months	M	83	100	78
Zhou [47]	2019	China	2009–2014	R	C	418 (209)	Yes	12 months	M	62	2	91

Abbreviations: NAR, Nerves at Risk; R, Retrospective; PNR, Prospective Non-Randomized; PR, Prospective Randomized; H, Historical; C, Contemporary; S, Single; M, Multiple; NS, Not Stated.

### 3.1. Meta-Analysis of All Studies Assessing Permanent Nerve Injury Categorized by NAR and per Patient

As shown in Figure 2, the incidence of permanent injuries when the unit of analysis was the NAR was 0.8% (549/67,887), corresponding to 0.69% (288/41,920) in the IONM group and 1.00% (261/25,967) in the visual identification only group. When assessed as a function of the number of patients, the incidence of permanent injuries was 1.50% (349/22,888), corresponding to 1.2% (144/11,639) in the IONM group and 1.8% (205/11,249) in the visual identification only group. (Figure 3) The OR of studies analyzed by NARs and per patient were 0.66 [95% CI 0.56 to 0.79; *p* < 0.00001] and 0.61 [95% CI 0.49 to 0.76; *p* < 0.0001], respectively, both favoring IONM.

### 3.2. Meta-Analysis of All Studies Assessing Total Nerve Injuries Categorized by NAR and per Patient

The total number of NARs in this subgroup was 46,596, of which 25,250 (54.2%) were in the IONM group and 21,346 (45.8%) in the visual identification group. The rates of total (permanent plus transient) RLN injuries assessed per NAR were 3.0% (774/25,250) in the IONM group and 4.2% (905/21,346) in the control group (OR 0.72; 95% CI [0.65, 0.79] (Figure 4). For studies analyzed per number of patients, there were a total of 26,058 patients. Those in the IONM group had a rate of 5.2% (695/13,294) total RLN injuries compared with 6.9% (887/12,764) in the control group (OR 0.71; 95% CI [0.64, 0.79]) (Figure 5). These data showed a statistically significant decrease in RLN injuries when using neuromonitoring intraoperatively, *p* < 0.00001 (NARs) and *p* ≤ 0.00001 (patients).

### 3.3. Meta-Analysis of Randomized Controlled Trials Assessing Permanent and Total Number of Nerve Injuries Categorized by NAR

Because randomized trials are the least likely to be biased, we analyzed those studies as a sub-analysis. Figure 6 displays a meta-analysis of permanent RLN injuries among a total of 4311 NARs in five RCT studies. In two of the five studies, there were no permanent injuries reported in both the IONM and non IONM groups. Figure 7 displays total RLN injuries which, in the IONM group, was 2.6% (56/2149 NARs) and 3.45% (75/2162 NARs) in the control group. In the five RCTs, we found no statistically significant benefit when using IONM compared to visualization alone in reducing the incidence of total (OR 0.87; [95% CI 0.52 to 1.45] *p* = 0.59) or permanent RLN injuries (OR 0.72; 95% [CI 0.32 to 1.64] *p* = 0.44). Although not statistically significant, the results strongly favor IONM.

### 3.4. Meta-Analysis of Studies with Documented Post-Operative Laryngoscopy Assessing Permanent and Total RLN Injuries Categorized by Nerves at Risk

Because the incidence of postoperative RLN injury depends upon the method of diagnosis of injury, and because postoperative laryngoscopy is the most secure way to diagnose postoperative RLN injury, we performed a subgroup analysis of studies employing postoperative laryngoscopy. We analyzed these studies only using NAR as the denominator. This yielded a statistically significant difference between using IONM (vs visualization alone) in reducing permanent (OR 0.67 (95% CI 0.55 to 0.80; *p* < 0.0001) (Figure 8)) as well as total RLN injuries (OR 0.68; 95% CI 0.61 to 0.76; *p* < 0.00001. Figure 9).

### 3.5. Meta-Analysis of All Studies with Contemporaneous Controls Assessing Permanent and Total RLN Injuries Categorized by Nerves at Risk

Because contemporaneous controls are considered more reliable than historical controls, we performed a subgroup analysis of the 36 studies employing contemporary controls.

As shown in Figure 10, the incidence of permanent RLN paralysis among these studies was 0.86% (452/52,073 NARs), corresponding to 0.72% (244/33,749) in the IONM group and 1.1% (208/18,324) in the control group. Figure 11 displays the results for total RLN injuries, 3.8% (1111/29,257 NARs) corresponding to 2.9% (469/16,148) in the IONM group and 4.9% (642/13,109) in the control group. These analyses proved to be statistically significant in favoring IONM in both groups: those with permanent injuries (OR 0.67; 95% CI [0.55, 0.82], *p* < 0.0001) as well as the total RLN injuries (OR 0.65; 95% CI [0.57, 0.74], *p* < 0.00001).

### 3.6. Meta-Analysis of All Studies with Contemporaneous Controls and Documented Postoperative Laryngoscopy Assessing Permanent and Total RLN Injuries Categorized by Nerves at Risk

Lastly, we performed an analysis of the studies we felt were most reliable, those with both contemporaneous controls and postoperative laryngoscopy. Among those reports, the incidence of permanent paralysis was 0.90% (429/47,329 NARs), corresponding to 0.77% (237/30,473) in the IONM group and 1.1% (192/16,856) in the control group (Figure 12). The total injuries were 4.0% (978/24,382 NARs), corresponding to 3.1% (398/12,825) in the IONM group and 5.0% (580/11,557) in the control group (Figure 13). Intraoperative neuromonitoring was associated with a significantly lower incidence of permanent injuries (OR 0.69; 95% CI [0.56, 0.84], *p* = 0.0003) and total RLN injuries (OR 0.63; 95% CI [0.55, 0.72], *p* < 0.00001).

Table 3 is a summary of the sub-group analyses displayed in the Figures above.

### 3.7. Risk of Bias Assessment

Employing the Cochrane Handbook for Systematic Reviews of Interventions [87], the bias domains assessed were (a) selection bias encompassing random sequence generation and allocation concealment, (b) performance bias to assess whether blinding of participants and personnel was undertaken, (c) detection bias (blinding of outcome assessment), (d) attrition bias (incomplete outcome data), and, finally, (e) reporting bias (selective reporting). Bias for each category was assigned a level of “high”, “low”, or “unclear” (Figure 14).

## 4. Discussion

Routine visual identification is considered the gold standard for the identification of the RLN to protect it from injury. The use of IONM has been proposed as a way to reduce the incidence of RLN injury. The use of IONM in thyroid surgery has reached approximately 50% in the United States and has approached 100% in Germany [86]. Although widely employed, debate continues regarding its efficacy and cost-effectiveness.

Several meta-analyses have been conducted in an attempt to resolve the debate. Our search of several databases identified 14 meta-analyses comparing the use of IONM with visual identification alone. Several of the meta-analyses found a decrease in both total and transient injuries in IONM cases, but for permanent injury, the results were particularly inconsistent.

Randomized studies would help resolve the issue, but because of the recent introduction of IONM to the field of thyroid and parathyroid surgeries, very few RCTs have been conducted to explore its benefits. Only five single-center, prospective RCTs have been conducted [28,40,45,64,69]. The study by Barczynski et al. [54], the largest with 2000 NARs, failed to find statistically significant results in reducing permanent RLN injuries. Similar results were obtained in the other four RCTs. It is of no surprise that our analysis of the RCTs failed to find a statistically significant benefit of using IONM to reduce total and persistent RLN injuries. Several reasons contributed to these findings. For one, the studies had small sample sizes which inherently lack the power to detect significant differences given the low incidence of RLN injuries. Sanabria [86] calculated that a sample size of 4500 patients or 9000 NAR would be required to demonstrate a statistically significant difference (ά 0.05, 80% power) for permanent injuries. For another, two of the five studies reported no permanent injuries in either the IONM or control patients, limiting statistical analysis. Third, four of the five RCTs assessed the performance of multiple surgeons. Because the performance of a single experienced surgeon is likely the most important determinant of postoperative vocal cord viability, particularly for high-risk surgeries, studies analyzing the performance of multiple surgeons could potentially affect the overall incidence of RLN injury. Finally, performance bias was seen in all five RCTs.

Sanabria et al. [88] assessed the methodologic quality of systematic reviews of IONM and highlighted issues that, in their judgment, compromised the reviews. These were (1) the underpowered nature of included studies, (2) failure to search a sufficient number of journal databases, (3) inclusion of studies that had no control group, (4) using a summary statistic, e.g., OR in place of an absolute estimator such as relative difference, (5) failure to report publication bias, and (6) using NAR as an analysis unit on the grounds that it artificially increases sample sizes. We have tried to address these issues as far as possible, considering that statisticians differ in their opinion regarding relative vs. absolute estimators. We analyzed data by both NAR and per patient. We feel NAR is the more clinically relevant analysis unit. Underlying this debate is the caveat that statistical difference is not equivalent to clinically relevant differences.

There are several sources of heterogeneity in the studies selected for our analysis. Differences in the control groups, historical vs. contemporaneous, are particularly important. Patients in historical control groups may lack baseline similarities with the treatment arm, resulting in confounding effects. In historical controls, patients might be selected from a pool of subjects that would favor the new treatment group, boosting the power of the trials at the cost of decreasing their generalizability. The quality of outcome information recorded for historical control records may differ substantially compared to contemporaneous control groups, since no study was underway requiring rigorous data acquisition at that time.

Another important source of heterogeneity is the definition of nerve injury and permanent nerve injury. Not all studies included laryngoscopy to assess vocal cord function; some relied upon subjective voice changes, which patients may have been reluctant to bring to the surgeons’ attention. Additionally, studies differed in the length of time before designating an injury as “permanent”.

Yet, other sources of heterogeneity, as reported in Table 2, include differing types of pathology, variable length of follow-up, varying proportions of men and women, and varying proportion of “high-risk” surgery in the selected studies.

## 5. Conclusions

A meta-analysis of all 60 studies demonstrated a statistically significant effect favoring the use of IONM in reducing the incidence of permanent RLN and total (transient and permanent) RLN injuries. Subgroup meta-analyses of studies considered the most reliable (those with routine postoperative laryngoscopy to define RLN injury, and those with contemporaneous, in contrast to historical, controls) also demonstrated statistically significant results favoring the use of IONM. Strong consideration should be given to employing IONM when performing thyroid surgery.

## Figures and Tables

**Figure 1 diagnostics-14-00860-f001:**
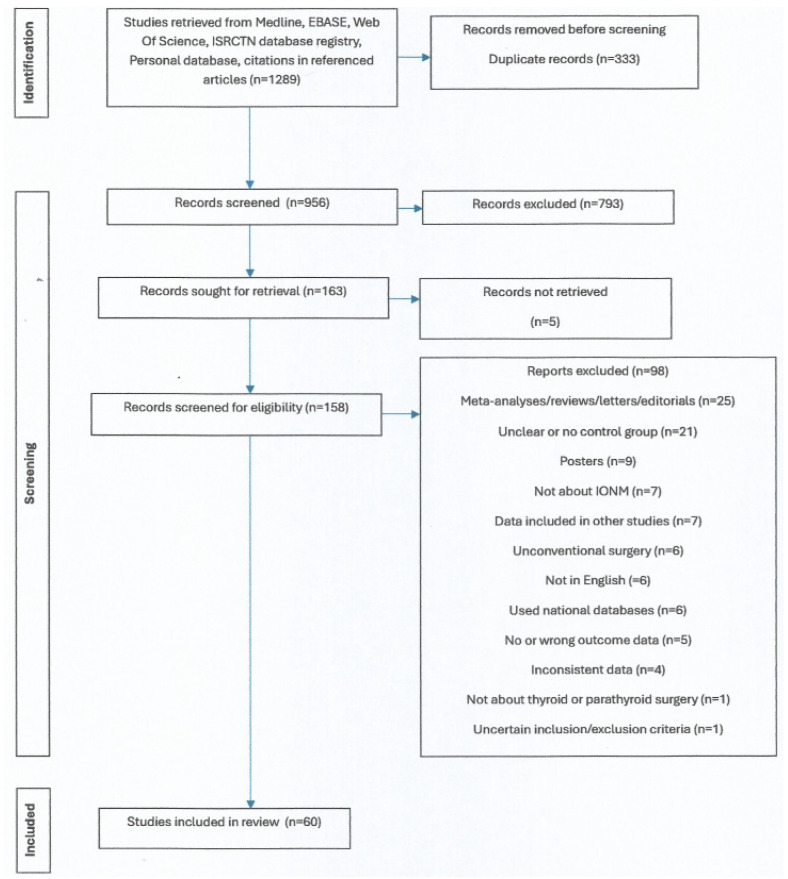
PRISMA flowchart of study identification and selection in the meta-analysis.

**Figure 2 diagnostics-14-00860-f002:**
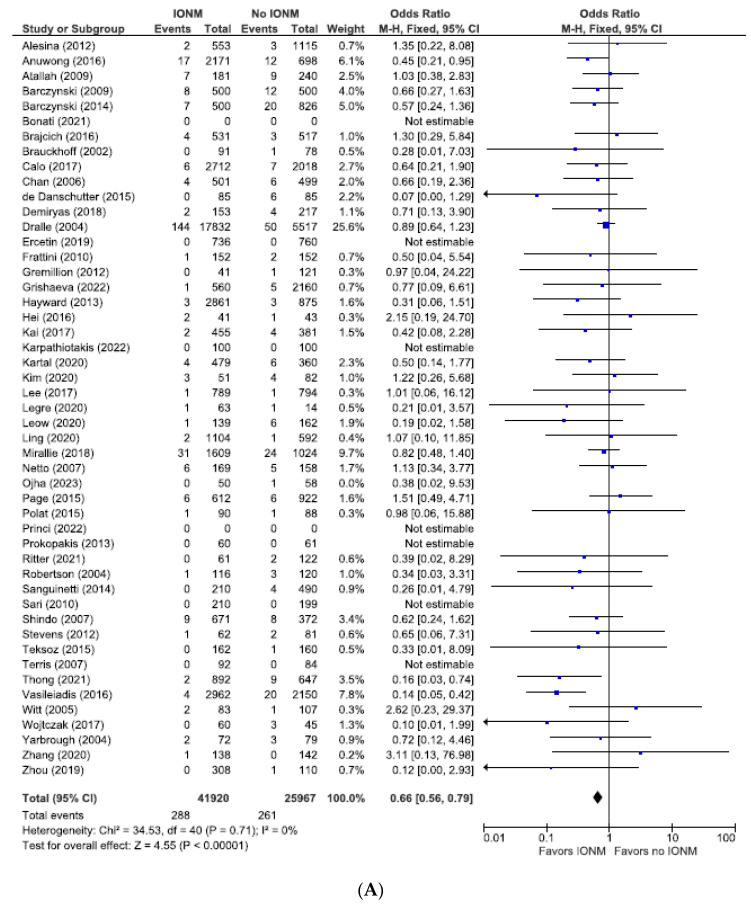
(**A**) Meta-analysis of all studies assessing permanent RLN injuries categorized by nerves at risk. (**B**) Funnel plot of heterogeneity [5,6,7,8,9,10,16,17,20,22,23,24,25,26,27,28,29,30,31,48,52,53,54,55,56,57,58,60,63,64,65,66,67,69,70,71,73,75,76,78,79,81,83,84,85,86].

**Figure 3 diagnostics-14-00860-f003:**
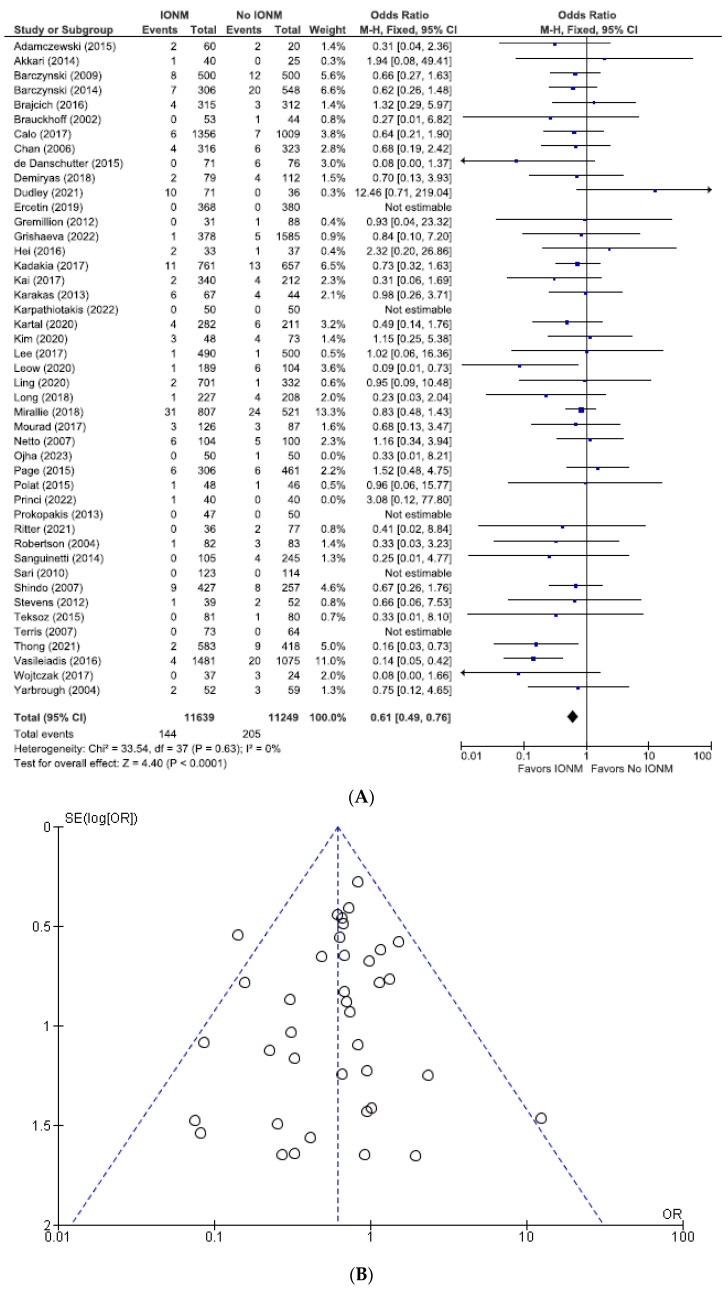
(**A**) Meta-analysis of all studies assessing permanent RLN injuries categorized by number of patients. (**B**) Funnel plot of heterogeneity [5,6,8,10,18,19,20,22,23,24,25,26,27,28,29,30,31,51,54,55,57,58,60,62,64,65,66,67,68,69,70,71,72,73,74,75,76,78,79,81,83,84].

**Figure 4 diagnostics-14-00860-f004:**
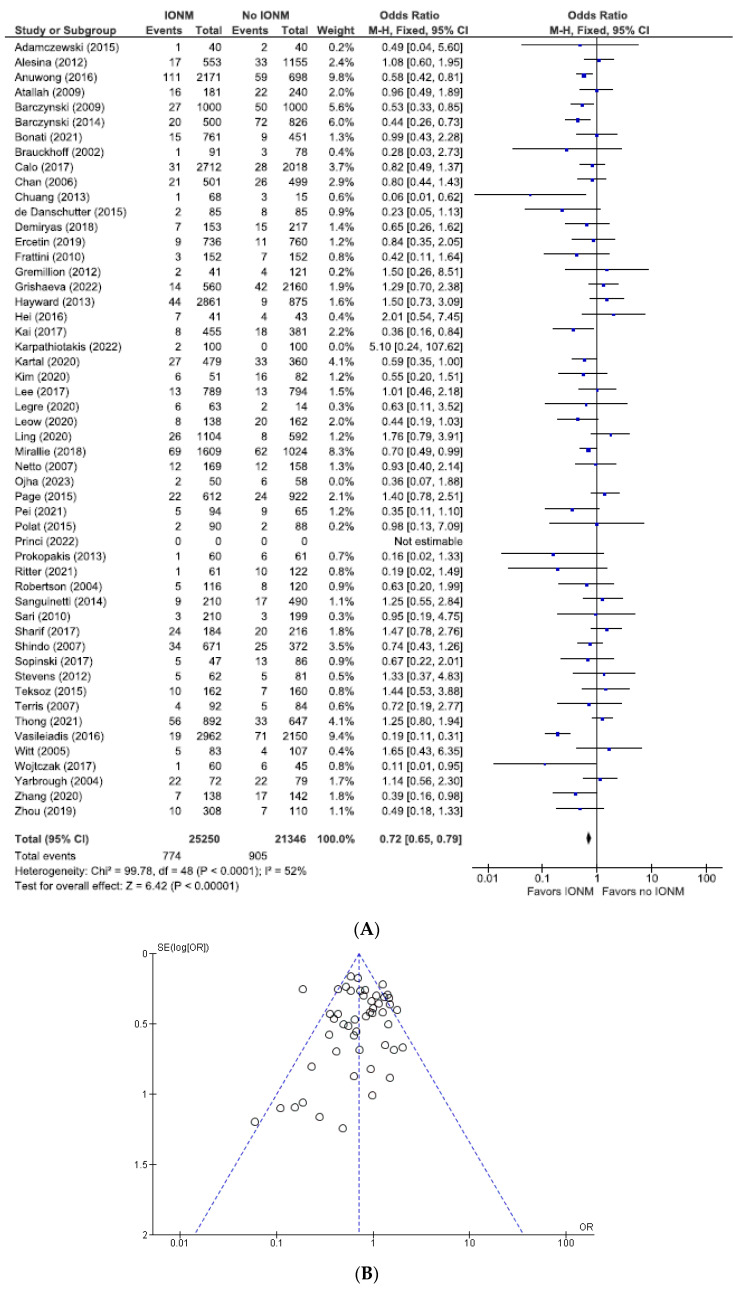
(**A**) Meta-analysis of all studies assessing total RLN injuries categorized by nerves at risk. (**B**) Funnel plot of heterogeneity [5,6,7,8,9,10,17,19,20,22,23,24,25,26,27,28,29,30,31,47,48,52,53,54,55,56,58,59,60,63,64,65,66,67,69,70,71,73,75,76,77,78,79,80,81,82,83,84,85].

**Figure 5 diagnostics-14-00860-f005:**
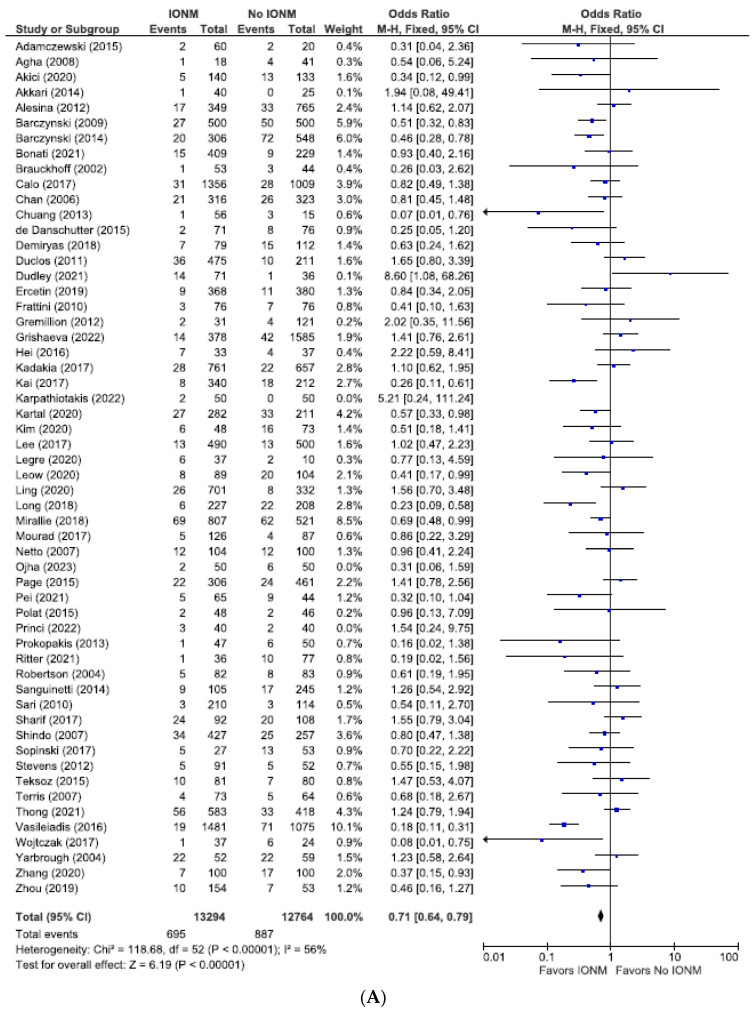
(**A**) Meta-analysis of all studies assessing total RLN injuries categorized number of patients. (**B**) Funnel plot of heterogeneity [5,6,7,8,10,17,18,19,20,22,23,24,25,26,27,28,29,30,31,47,49,50,51,52,54,55,56,58,59,60,61,62,63,64,65,66,67,69,70,71,72,73,74,75,76,77,78,79,80,81,82,83,84].

**Figure 6 diagnostics-14-00860-f006:**
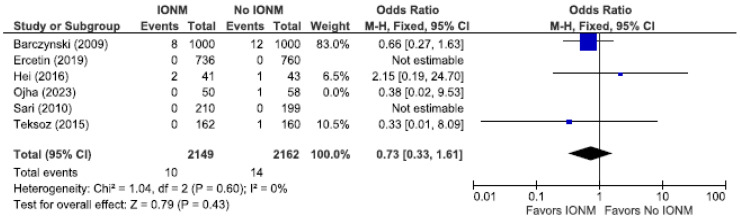
Meta-analysis of randomized studies assessing permanent RLN injuries categorized by nerves at risk [25,27,29,54,66].

**Figure 7 diagnostics-14-00860-f007:**
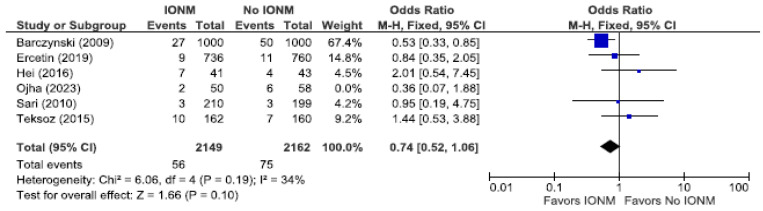
Meta-analysis of randomized studies assessing total RLN injuries categorized by nerves at risk [25,27,29,54,66].

**Figure 8 diagnostics-14-00860-f008:**
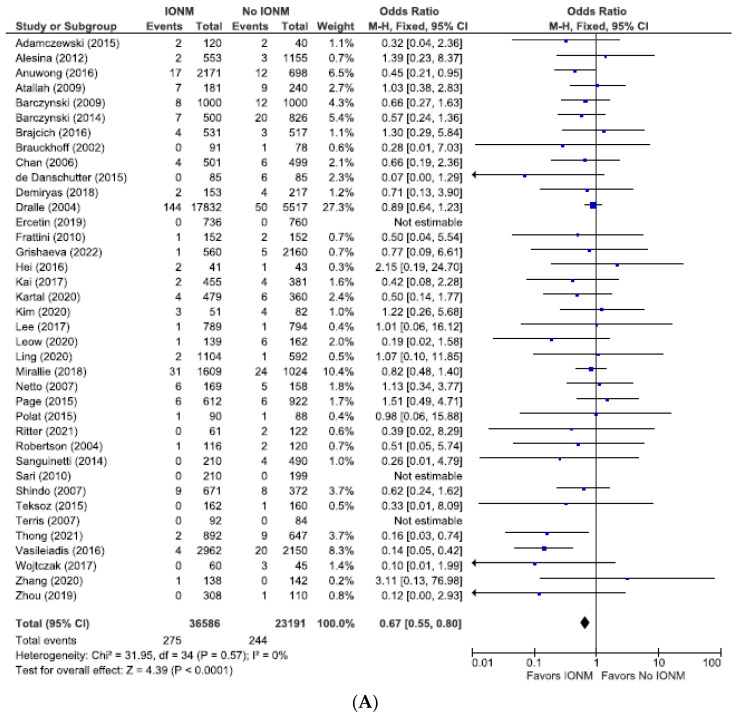
(**A**) Meta-analysis of permanent RLN injuries among studies with post-operative laryngoscopy categorized by nerves at risk. (**B**) Funnel plot of heterogeneity [6,7,8,10,16,19,20,23,24,25,26,27,28,29,31,48,52,53,54,55,57,58,60,63,65,66,67,69,70,71,73,75,76,78,79,81,84,86].

**Figure 9 diagnostics-14-00860-f009:**
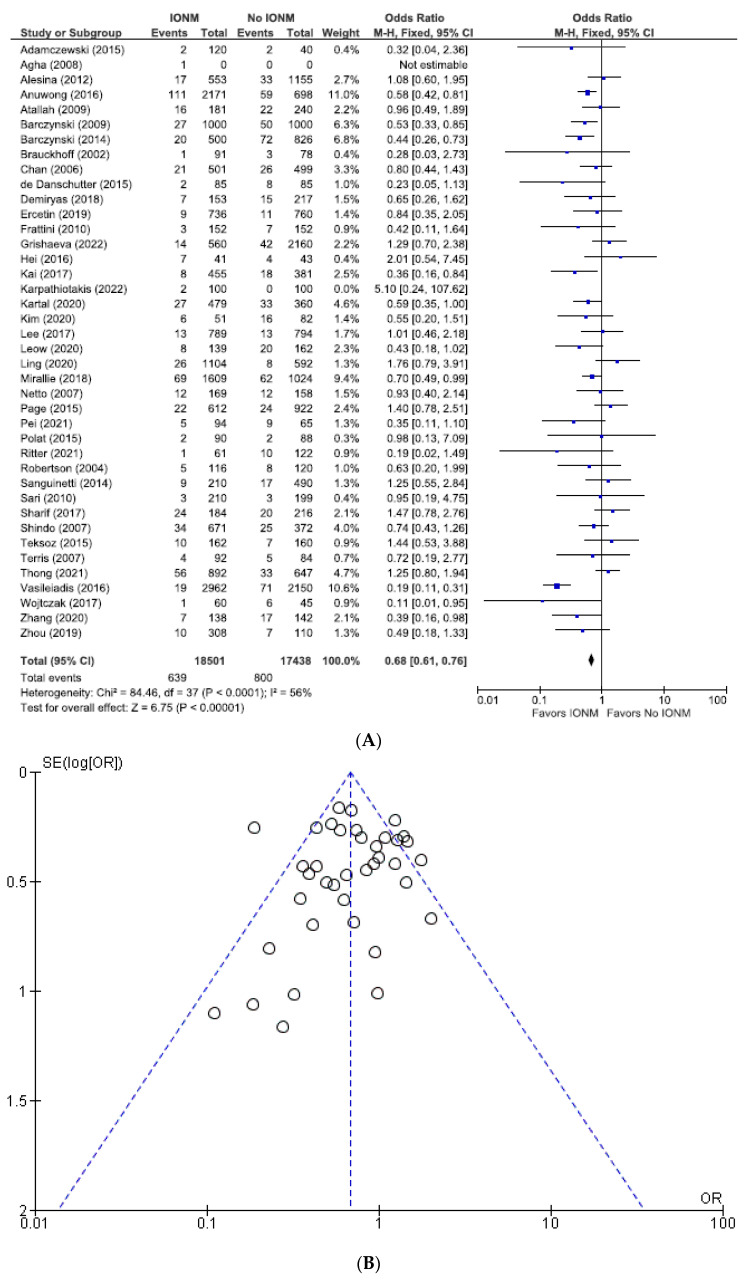
(**A**) Meta-analysis of total RLN injuries among studies with post-operative laryngoscopy categorized by nerves at risk. (**B**) Funnel plot of heterogeneity [6,7,8,10,19,20,23,24,25,26,27,28,29,31,47,48,49,52,53,54,55,58,60,63,65,66,67,69,70,71,73,75,76,77,78,79,80,81,84].

**Figure 10 diagnostics-14-00860-f010:**
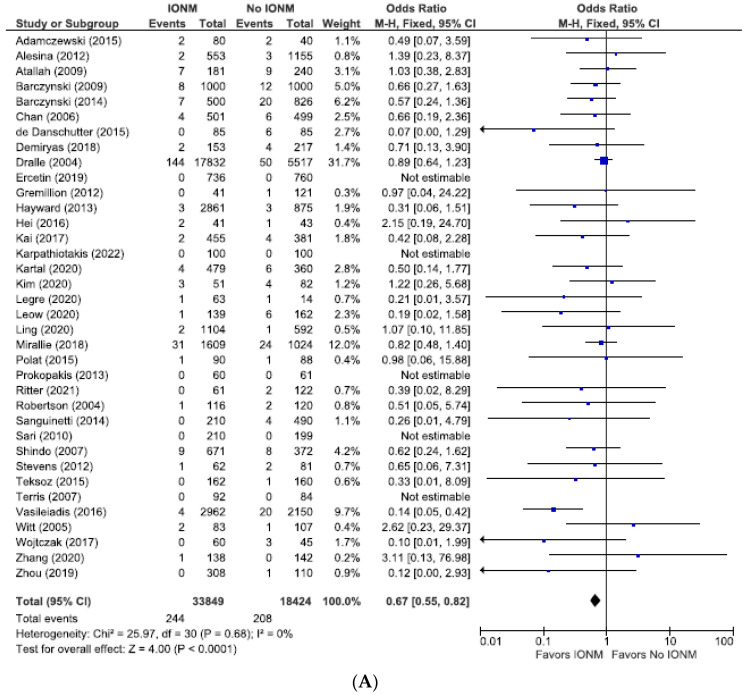
(**A**) Meta-analysis of permanent RLN injuries among studies with contemporaneous controls categorized by nerves at risk. (**B**) Funnel plot of heterogeneity [6,7,8,9,10,16,17,19,20,24,25,26,27,28,29,30,31,47,52,53,54,55,58,60,64,66,67,70,71,73,78,79,81,83,85].

**Figure 11 diagnostics-14-00860-f011:**
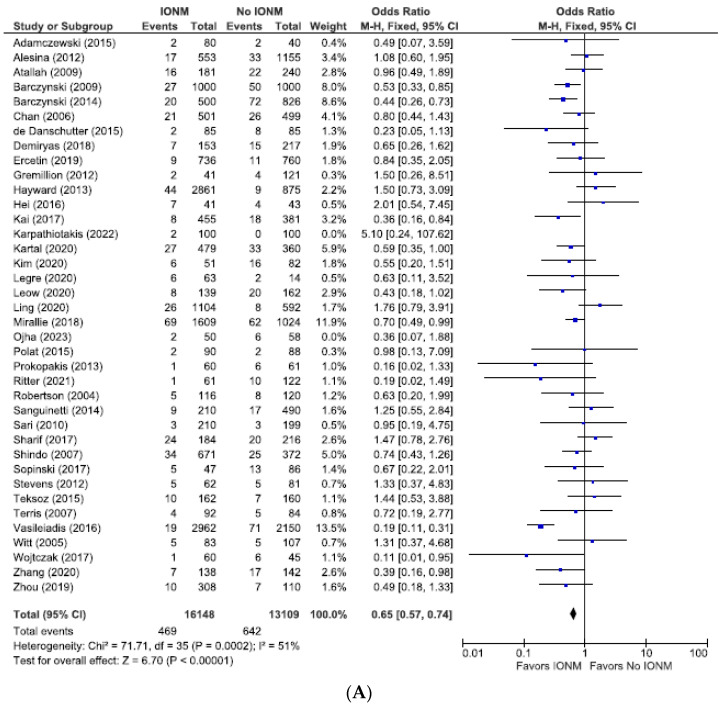
(**A**) Meta-analysis of total RLN injuries among studies with contemporaneous controls categorized by nerves at risk. (**B**) Funnel plot of heterogeneity [6,7,8,9,10,17,19,20,24,25,26,27,28,29,30,31,47,52,53,54,55,58,60,64,66,67,70,71,73,78,79,80,81,82,83,85].

**Figure 12 diagnostics-14-00860-f012:**
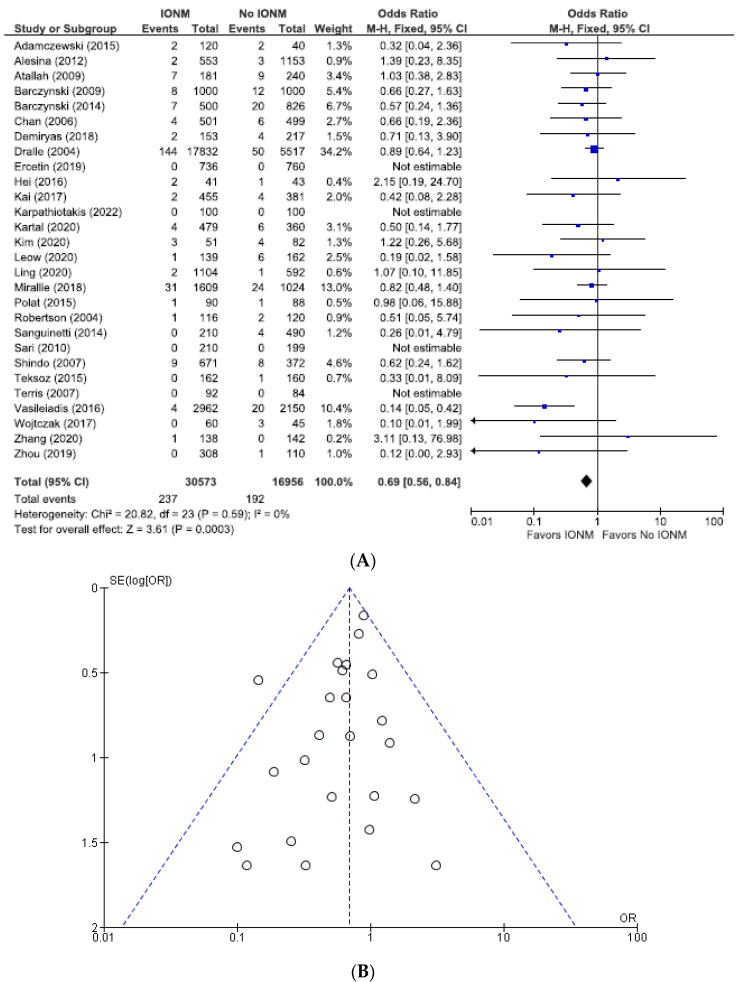
(**A**) Meta-analysis of permanent RLN injuries among studies with contemporaneous controls and post-operative laryngoscopy categorized by nerves at risk. (**B**) Funnel plot of heterogeneity [6,7,8,10,16,19,20,25,27,28,29,31,47,48,52,54,55,58,60,66,67,70,71,73,78,79,81].

**Figure 13 diagnostics-14-00860-f013:**
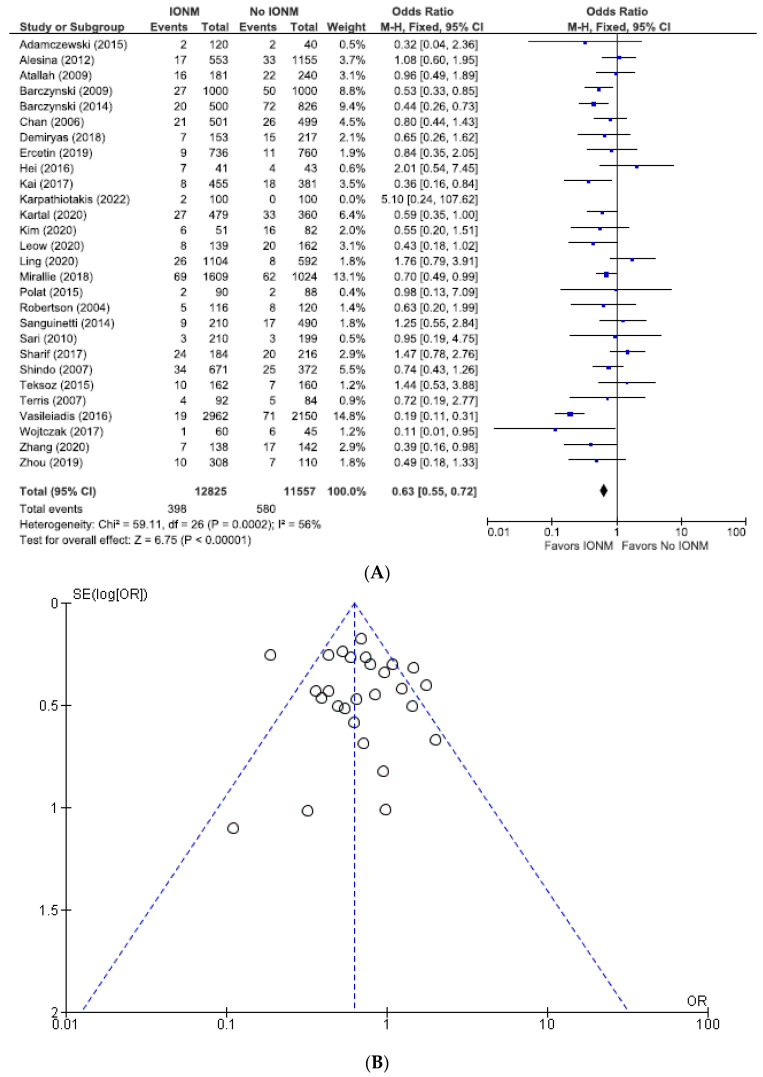
(**A**) Meta-analysis of total RLN injuries among studies with contemporaneous controls and post-operative laryngoscopy categorized by nerves at risk. (**B**) Funnel plot of heterogeneity [6,7,8,10,19,20,25,27,28,29,31,47,52,53,54,55,58,60,66,67,70,71,73,78,79,80,81].

**Figure 14 diagnostics-14-00860-f014:**
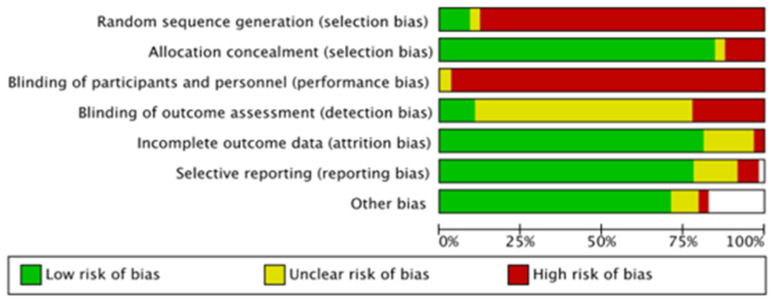
Risk of bias among studies in this meta-analysis.

**Table 1 diagnostics-14-00860-t001:** Meta-analyses of intraoperative nerve monitoring.

Author	Year	Favors IONM in Reducing Permanent RLN Injuries	Favors IONM in Reducing Transient RLN Injuries	Favors IONM in Reducing Total RLN Injuries
Bai [32]	2018	Yes	Yes	Yes
Cleere [33]	2022	No	No	NS
Cirocchi [34]	2019	No	No	No
Davey [35]	2022	No	No	No
Higgins [36]	2011	No	No	No
Kim [37]	2021	Yes	Yes	Yes
Lombardi [38]	2016	No	NS	NS
Pisanu [39]	2014	No	No	No
Rulli [40]	2014	No	Yes	NS
Sanabria [41]	2013	No	No	NS
Sun (reoperations) [42]	2017	Yes	No	Yes
Wong (high-risk patients) (Random effects analysis) [43]	2017	No	No	Yes
Wong (high-risk patients) (Fixed effects analysis) [43]	2017	No	Yes	Yes
Yang [44]	2017	No	No	No
Zheng [45]	2013	No	Yes	Yes

**Table 3 diagnostics-14-00860-t003:** Summary of sub-group meta-analyses.

Category	Type of Injury	Assessment	% Injury IONM	% Injury No IONM	OR (95% CI) IONM vs. No IONM
All studies	Permanent	NAR	0.69	1.0	0.66 (0.56–0.79)
		Patient	1.2	1.8	0.61 (0.49–0.76)
All studies	Total injuries	NAR	3.0	4.2	0.72 (0.65–0.79)
		Patient	5.2	6.9	0.71 (0.64–0.79)
Randomized	Permanent	NAR	0.47	0.65	0.73 (0.33–1.61)
	Total	NAR	2.6	3.4	0.74 (0.52–1.06)
Postoperative laryngoscopy	Permanent	NAR	0.75	1.0	0.67 (0.55–0.80)
	Total	NAR	3.4	4.6	0.68 (0.61–0.76)
Contemporaneous controls	Permanent	NAR	0.72	1.1	0.67 (0.55–0.82)
	Total	NAR	2.9	4.9	0.65 (0.57–0.74)
Contemporaneous controls AND laryngoscopy	Permanent	NAR	0.78	1.1	0.69 (0.56–0.84)
	Total	NAR	3.1	5.0	0.63 (0.55–0.72)

NAR: analyzed by number of nerves at risk. Patient: analyzed by number of patients. IONM: intraoperative nerve monitoring. OR: odds ratio. OR less than 1.0 favors IONM; 95% CI: 95% confidence interval.

## Data Availability

The data presented in the study are openly available in PubMed, EMBASE, Cochrane, Web of Science, and ISRCTN registry databases.

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
