# Peer review of "Does the Use of Intraoperative Neuromonitoring during Thyroid and Parathyroid Surgery Reduce the Incidence of Recurrent Laryngeal Nerve Injuries? A Systematic Review and Meta-Analysis"

_diagnostics, 2024, doi:10.3390/diagnostics14090860_

Round 1
Reviewer 1 Report
Comments and Suggestions for Authors
The study would achieve greater relevance if only prospective randomised trials with pre operative and post operative laryngoscopy for diagnosis of nerve injury and clear universally accepted definiton of "temporary nerve palsy ' and "permanent nerve palsy" were included .Suggestions for criteria for further studies in the subject should be included - such as prospective randomised trials with contemperous controls and clear definition criteria of iatrogenic nerve paralysis- permanent and temporary with follow up
Author Response
The authors thank Reviewer #1 for his/her thoughtful comments.
Indeed, it would be instructive to be able to analyze studies that reported outcomes of temporary nerve palsy using standardized definitions. Unfortunately, as displayed in Table 2, there is no universally accepted definition. Among the studies we included, as many employed a definition of 6 months as 12 months. If the editor feels that an additional sub-analysis of studies employing various definitions is important, we can perform such an analysis, however considering that temporary nerve palsies were not the main focus, or the most clinically important, such an analysis may not be of great help.
The suggestion that we discuss plans for future studies is intriguing. Certainly, that is a part of reporting upon original research. In the case of systematic reviews and meta-analyses, future plans can only consist of another review after additional studies have been published
Reviewer 2 Report
Comments and Suggestions for Authors
The authors showed usefulness of IONM during thyroid and parathyroid surgery in reducing RLN by meta-analysis. The content seemed reliable and results should be interesting for readers.
Author Response
The authors thank Reviewer #2 for his/her supportive comments.
Reviewer 3 Report
Comments and Suggestions for Authors
This manuscript is a meta-analysis of studies that investigated the effectiveness of intraoperative recurrent laryngeal nerve monitoring in reducing the incidence of recurrent laryngeal nerve injury during thyroid and parathyroid surgery. The study is well designed and the text is clearly written. There is a great inconsistency in the scientific literature regarding this topic, so I consider the results of this study very valuable and important.
I recommend to accept this manuscript in the present form.
Author Response
The authors thank Reviewer #3 for his/her supportive comments.